# Soil Degradation Mapping in Drylands Using Unmanned Aerial Vehicle (UAV) Data

**Juliane Krenz *** , **Philip Greenwood and Nikolaus J. Kuhn**

Physical Geography and Environmental Change, University of Basel, 4056 Basel, Switzerland;
Philip.greenwood@unibas.ch (P.G.); nikolaus.kuhn@unibas.ch (N.J.K.)
*** Correspondence: juliane.krenz@unibas.ch

**Abstract:** Arid and semi-arid landscapes often show a patchwork of bare and vegetated spaces. Their heterogeneous patterns can be of natural origin, but may also indicate soil degradation. This study investigates the use of unmanned aerial vehicle (UAV) imagery to identify the degradation status of soils, based on the hypothesis that vegetation cover can be used as a proxy for estimating the soils' health status. To assess the quality of the UAV-derived products, we compare a conventional field-derived map (FM) with two modelled maps based on (i) vegetation cover (RGB map), and (ii) vegetation cover, topographic information, and a flow accumulation analysis (RGB+DEM map). All methods were able to identify areas of soil degradation but differed in the extent of classified soil degradation, with the RGB map classifying the least amount as degraded. The RGB+DEM map classified 12% more as degraded than the FM, due to the wider perspective of the UAV compared to conventional field mapping. Overall, conventional UAVs provide a valuable tool for soil mapping in heterogeneous landscapes where manual field sampling is very time consuming. Additionally, the UAVs' planform view from a bird's-eye perspective can overcome the limited view from the surveyors' (ground-based) vantage point.

**Keywords:** erosion; landscape mapping; soil degradation; soil mapping; unmanned aerial vehicle (UAV)

## 1. Introduction

Arid and semi-arid landscapes often show a heterogeneous pattern of bare and vegetated spaces [1–4]. Most common vegetation patterns are banded [5,6] or spotted [1,7]. This patchwork of vegetation is also reflected in heterogeneous chemical, structural, and textural soil properties [8,9], which introduce spatial variations in factors such as infiltration capacity [10–12], soil nutrient content [8,13] and soil erodibility [12,14,15]. The vegetation patterns can be of natural origin [16,17], but may also indicate soil degradation, i.e., the decline of soil functions and productivity caused by erosion, nutrient depletion, salinization, or loss of soil structure [10].

The spatial heterogeneity of dryland landscapes indicates that underlying soil types are inherently variable. This variability is often not depicted on soil maps. Most common soil maps for South Africa are large scale, varying between 1:1 and 1:5 million and including the Soil and Terrain (SOTER) database [18] and the "Soil Atlas of Africa" [19], and are not able to represent the variety of the natural soil types. Ref. [20] used random forest models to predict common soil properties, such as organic carbon, pH, and soil texture, at a 250 m × 250 m resolution to bridge the gap in soil information in Africa. Although a large amount of soil investigations has been carried out in South Africa, information access is restricted. Local or provincial soil information is often archived and not freely available to the public due to a lack of a central repository [21]. Long-term erosion monitoring in the Karoo by [22] showed evidence of considerable soil degradation and badlands formation but information on the soils' degradation status and the loss of soil functions and productivity is usually not included in soil maps. In addition to this limited access, national or regional maps seldom reflect this degree of heterogeneity, mainly due to the high labor, financial, and time-investment costs associated with manual

land cover mapping, soil sampling, and analysis. However, high resolution soil maps are of growing importance for environmental planning at regional or local scales [23] and for farmers to facilitate decision-making on best agricultural practices [24], such as precision farming or grazing practices aimed at land restoration [20], as well as for conservationists to implement protection measures in the form of erosion prevention [20,24]. Digital soil mapping using mathematical models therefore represents an alternative to the typically resource-intensive conventional methods of soil mapping [25]. Soil characteristics or soil types are usually predicted based on factors about the terrain or climate [26], using classical machine learning methods such as classification and regression tree analyses [26,27], neural networks [23], or geostatistical approaches [28]. But since vegetation cover is closely related to soil properties [8], land cover maps can provide useful information to identify differences in soil development [29].

The quality of the prediction produced by any soil mapping model strongly depends on the spatial scale of the input data. The spatial resolution of commonly available global or continental data products on land cover, such as open source Global Land Survey (GLS) data from NASA and the US Geological Survey (30 m × 30 m), MODIS-based Global Land Cover Climatology (500 m × 500 m) [30], or soil types such as the African Soil Atlas [31], is usually too coarse to reflect landscape heterogeneity in semi-arid areas. Unmanned aerial vehicles (UAVs) can help to overcome the gap between expensive, time-consuming ground-based assessment and insufficient data quality from coarse resolution imagery derived from satellite systems. Capturing high resolution terrain or land cover information enables the generation of user-specific data products, such as 2D or 3D terrain models, orthomosaics, and normalized difference vegetation index (NDVI)-maps. Consequently, UAVs have recently gained popularity in remote sensing studies and have been used in a variety of high resolution topographic studies, e.g., for gully mapping [32] and quantifying gully volumes [33–35]. Compared to field mapping, which produces rather inflexible maps once compiled and the results committed to paper, or a similarly static digital data products due to the intense labor involved in their creation, UAV mapping can be conducted more frequently at lower cost and with a finer resolution, which allows rapid monitoring of changes in natural soil states such as displacement of soil after landslides [36] or monitoring the evolution of an active volcano [37]. UAV imagery does provide limited information on the soils themselves but can be used to infer, for example, the state of soil degradation through density and patterns of vegetation or through capturing fine-scaled features of topography such as rills and gullies.

Based on the relationship between the state of a soil and its vegetation cover [38,39], this study investigates the use of UAV imagery to identify soil degradation. In addition, intense destruction of soil by erosion can be associated with erosion features, such as rills or badlands, visible in digital terrain models [40,41]. Data generated by inexpensive and commercially available UAVs were chosen on purpose because they offer the greatest potential for wide application in soil and land management. To assess the quality of these UAV-derived products on vegetation and topography, we compare a conventionally derived field map with two modelled maps. One is exclusively based on vegetation cover, assuming vegetation is indicative of low soil degradation by erosion [40,41]. The second map is based on vegetation cover combined with terrain information, assuming that visualization of features such as rills, gullies, and sediment deposits contribute to assessing soil degradation and redistribution.

## 2. Materials and Methods

### 2.1. Study Site

The chosen study area is located in the Klein Seekoei River basin in the Great Karoo Region of South Africa and is situated approximately 70 km north of Graaff-Reinet. It is part of the Sneeuberg uplands, a landscape of flat valley bottoms and gently sloping lower valley sides interspersed with hills and mountain ranges extending from 1650 up to 2502 m above sea level (a.s.l) at the Compassberg. According to [42], in areas of the Karoo with seasonally distributed precipitation of almost 500 mm per year, such as the Sneeuberg uplands, in the absence of human intervention a vegetation cover of 50–70% could be expected. Land degradation is a common problem in the area and is reflected by

frequent rill and gully erosion in the valley bottoms, as well as badland development on the footslopes of many hills [22,43–45]. The study catchment covers 3.2 km$^2$ and drains into the in-filled reservoir labelled Dam 53 (31.698558° S, 24.588183° E) by [46]. The reservoir is located at approximately 1740 m a.s.l. and is surrounded by hills up to 2080 m a.s.l. Aerial photos indicate that it was constructed in the late 1920s and almost completely filled with sediment by the mid-1970s. A main gully up to 6 m deep and partly up to 10 m wide follows the thalweg of the catchment, forming a breach in the dam wall and incising backwards into the sediment. We focused on the lower part of the catchment, just behind the former dam wall, which shows various grades of land degradation and covers roughly 289 ha. The catchment was chosen because it shows many typical features of vegetation degradation, erosion and deposition of soil. The relatively small size meant that detailed mapping was feasible, both on the ground as well as with the UAV.

## 2.2. Field Mapping

In an on-site assessment throughout the complete study area, differences in vegetation cover or vegetation type, as well as prominent erosion features (i.e., rills, gullies, and extended absence of A-horizon), were visually assessed and their positions were recorded with a handheld Garmin GPSMap60. The GPS data of the recorded land cover types and erosion features was transferred to a GIS and transformed into the Field Map (FM). Six different types of vegetation cover and erosion could be distinguished: shrubs, thorny shrubs, grasses, mixed vegetation, bare soil, and erosion rills. Detailed descriptions of selection criteria and distinguished vegetation and erosion types are presented in Table 1.

## 2.3. Land Cover Mapping

To create a high resolution map of soil degradation in our study area, a land cover classification was carried out using UAV imagery acquired in February 2016 using a Phantom III Professional quadrocopter (SZ DJI Technology Co., Ltd., Shenzhen, China) equipped with a DJI FC300X camera (GoPro Inc., San Mateo, CA, USA) for RBG-images according to [47]. Images were taken at an altitude of 70 m above the ground with an average overlap of 90% to ensure sufficient overlap for photogrammetric processing. Compromising between a high resolution and the expenditure of image acquisition and post-processing resulted in a ground size resolution of 3.1 cm of the orthomosaic. Image processing, including orthorectification and production of a high-resolution orthophoto and a digital elevation model (DEM), was done with the photogrammetry software Pix4Dmapper Pro (Pix4D SA, Lausanne, Switzerland). The land cover classification map generated by the supervised classification of the orthomosaic is referred to as the Land Cover (LC). In our study, land cover only refers to the actual surface cover, such as grasses, shrubs, stones or bare soil, and disregards any additional information on the relief. Even though we mapped six different classes on site for the FM, we combined those classes as shown in Table 1 to simplify the land cover classification.

The LC was classified using a supervised support vector machine (SVM) classification with the help of ArcGIS Pro 2.2.3. This classification was chosen over the very popular maximum likelihood classification (MLC) approach because it has shown more accurate results in other studies [48–50]. The pixel-based MLC approach assigns a pixel to the corresponding class with the maximum likelihood. It is a parametric classification that is limited by assuming a normal distribution of class signatures which requires the users to determine the classification scheme. Unlike the MLC, the non-parametric SVM classification employs optimization algorithms to locate optimal boundaries between classes [48]. While its strength lies in a binary classification system, the SVM has recently also been shown to reflect accurate land cover categories in multiclass approaches [50,51]. An image analysis pre-study was conducted to identify the segmentation parameters that yielded in the most accurate segmentation of the orthophoto. Forty training samples per LC class (bare soil, shrubs, and grasses) were used for the LC classification and each segmented image was tested for accuracy as described in 2.4. We found the parameters producing the most accurate segmented image to be 20 (spectral detail), 15 (spatial detail), and 5 (minimum segment size in pixels) and used the resulting image for the SVM analysis.

**Table 1.** Description of the identified land cover types. The vegetation cover (first column) describes classes visually assessed in the field, while the LC class describes the classification used for the image analysis of the orthophoto. The arrows indicate how individual vegetation cover classes were combined to give LC classifications. Descriptions were based on those given in [52] and have been modified according to site-characteristics.

| Vegetation Cover | Description | Implication | Land Cover (LC) | Identification from the Orthophoto |
|---|---|---|---|---|
| Bare soil | Areas (>1 m$^2$) of exposed sand, soil, or rock with no or very little vegetation | Threatened by erosion due to lack of protecting vegetation cover | Bare Soil | Higher overall reflectance (whiter appearance) → areas of mostly white to sandy colors (often slightly yellow or orange) with no apparent line structure that could indicate grasses; stones often very white and characterized through their angular shape |
| Rills | Rill structure of bare soil, no or little shrub vegetation | Threatened by intensive erosion due to concentrated runoff | | |
| | Woody-plants dominating, typically broad-leaved, branching at or near the ground, up to 1 m in height | Threatened by erosion if roots are exposed or shrubs grow on pedestals | Shrubs | In this area characterized by a light bright green to dark green or blueish green color |
| Thorny Shrubs | Woody-plants with thorns dominating, up to 1 m in height, mostly *Lycium horridum* | Threatened by erosion if roots are exposed or shrubs grow on pedestals | | |
| Mixed Vegetation | Areas where neither grasses nor shrubs are dominant | Threatened by erosion if vegetation cover <50% | | |
| Grasses | Non-woody, grass-like, herbaceous plants dominating, grass-like plants often grow as tussocks | Threatened by erosion if vegetation cover <50% or tussock grasses grow on pedestals | Grasses | Color appearance is more faded (beige to light grey green) than the color of the shrubs; in a high resolution image grasses can be distinguished from shrubs by their long, narrow, and sometimes curled leaves |

### 2.4. Accuracy Assessment

To assess the LC map accuracy, and thus compare the LC classes expressed in the UAV-derived map to reference data collected, 150 randomly stratified points per LC class were generated on the classified LC map. These were then validated visually point by point using the orthophoto and field knowledge to create a classified point dataset, which was used for further accuracy analysis. A confusion matrix representing the overall accuracy, producer's and user's accuracy, and the kappa index [53,54] was calculated using ArcGIS. The kappa index is a measure for the accuracy of the classified map compared to the reference map. It is computed as follows

$$K = \frac{N \sum_{i=1}^{n} m_{i,i} - \sum_{i=1}^{n}(R_i C_i)}{N^2 - \sum_{i=1}^{n}(R_i C_i)} \tag{1}$$

Definitions for the variables used in Equation (1) may be given as:

$i$—class number
$N$—total number of classified values compared to reference values
$m_{i,i}$—number of values belonging to reference class I that have also been classified as class $i$
$C_i$—total number of predicted (classified) values belonging to class $i$
$R$—total number of reference values belonging to class $i$

Producer's accuracy refers to the probability that land cover on the ground is classified as is given on the map. It is calculated by dividing the number of correctly classified reference points by the total number of reference points for a land cover class. User's accuracy, on the other hand, indicates how often a class on the map will actually be present on the ground. For user's accuracy the total amount of correct classifications for a particular class is divided by the total number of sampling points of the referring LC class, which in our case was 150.

### 2.5. Landscape Unit (LU) Mapping

The second map generated from the UAV data is referred to as the Landscape Unit (LU) map because it also contains information on topography. The rationale for distinguishing landscape units is the relationship of soil, soil erosion, and deposition to terrain. Areas of interest were, in particular, bare soil areas with visible erosion features such as rills and areas with exposed shrub roots, since these are indicators of erosion and soil degradation [55]. Vegetation cover greater than 50% was used as an index for healthy soil [42]. A transformation of the Land Cover (LC) raster to a LC density map to assess the percentage of bare soil cover was needed. Therefore, the LC raster was converted to a point dataset, where each raster cell represented a point with its specific land cover (bare soil, shrub or grass). Using a Kernel density function the density of bare soil points in the neighbourhood was. Two bare soil threshold maps were generated from the Kernel density map: (i) bare soil larger than 50% representing moderately degraded areas and (ii) bare soil larger than 70% representing severely degraded areas. Areas smaller than 10 m$^2$ and fully enclosed by an area of different degradation status were appointed to the surrounding area. We assumed that the different classification of these tiny patches was mainly attributed to uncertainties in a distinct LC class allocation of transitional areas and therefore allocated these areas to the surrounding LU. For the final LU maps we used two approaches (Figure 1, Figure 2. For the RGB map, which is based on land cover, the orthophoto (Figure 4A), the LC map (Figure 4B), and the derived bare soil density map were exclusively used. For simplification and because the map is based on a true colour image (the orthophoto), we named it RGB map based on the additive primary colours red, green and blue that are typically used for the display of images in digital cameras. For the RGB+DEM map, LUs were categorized based on visual and elevation differences combining information from the orthophoto, the LC map, the density map, the high-resolution DEM (Figure 4C), and the DEM derivatives, such as slope and flow direction. The extent of the different LUs was then mapped using an overlay analysis in ArcGIS. The depositional area was in both approaches

mapped based on field observations and historical aerial imagery that showed the spatial extent of the former reservoir.

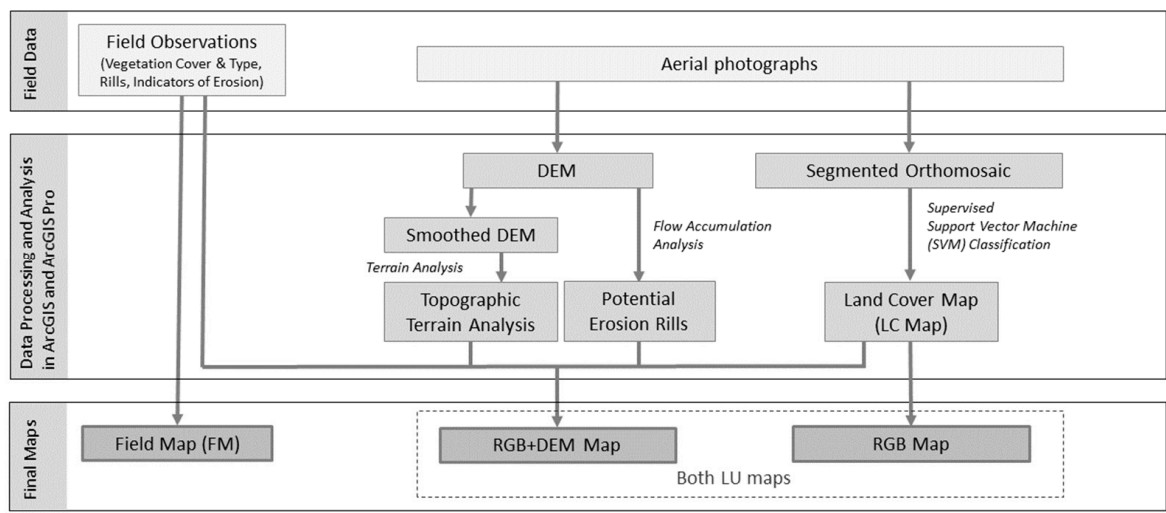

**Figure 1.** Workflow for image classification and landscape unit mapping.

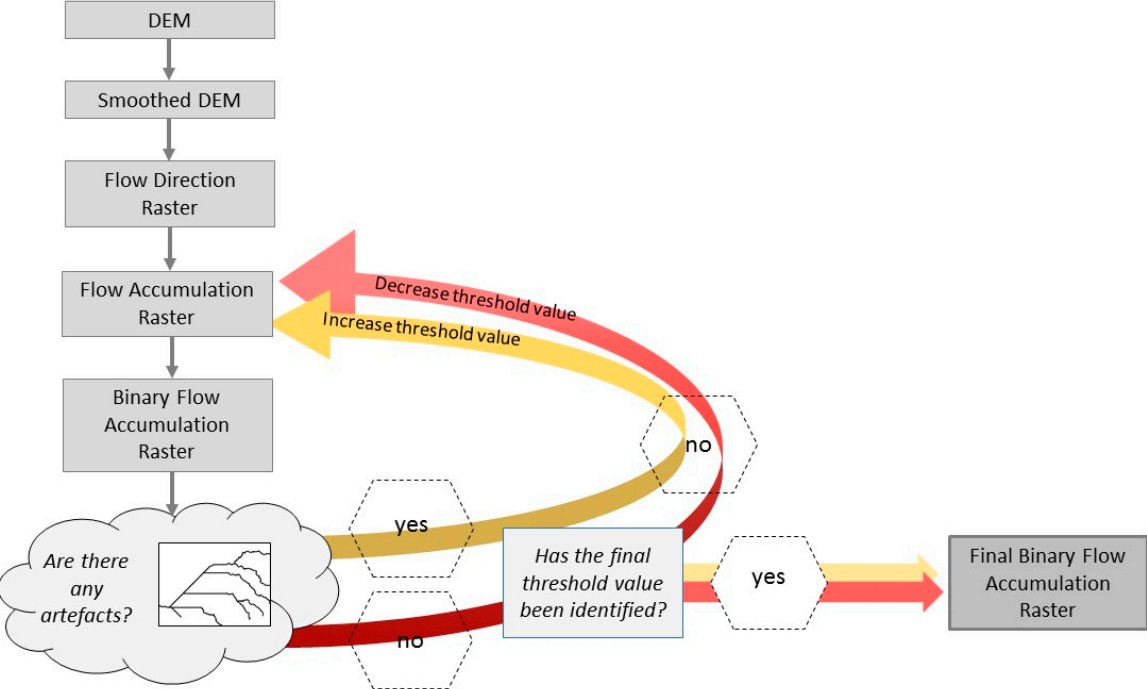

**Figure 2.** Workflow of the flow accumulation analysis for the identification of potential erosion rills.

### 2.6. Incorporating Terrain Attributes in LU Mapping

Soils and landscapes co-evolve and are inherently related to each other [56,57]. Therefore, we used the DEM and DEM derivatives to identify areas of similar states of soil degradation. The most frequent terrain attributes used for digital soil mapping are elevation, slope, aspect, and curvature, often combined with hydrological features, such as flow accumulation [23]. Aspect was found to have a low predictive value for estimating soil attributes by [58]. Since our study area is fairly small and differences in aspect or curvature of the foot slope are small, we decided to focus on flow direction, flow accumulation, and the topographic position index (TPI).

Flow accumulation analysis of DEMs can be used to identify rills which are indicative of erosion and/or transport of sediment from an eroding upslope area [59]. To test the quality of this analysis, the result was compared to actual rills mapped in the field. An unweighted (accumulation with equal cell weights) single-flow path model was used in ArcGIS to identify flow accumulation paths in the study area (Figure 2). Identified paths were then compared with on-site observations of erosion rills. A single-flow path model was chosen because it identifies parallel and converging water and sediment fluxes, which is ideal to study connected rill systems and mass fluxes that are trapped and stored behind dams [60]. The resulting image from the initial flow accumulation analyses was reclassified using a threshold value to highlight the pathways of high flow and to create a stream network raster, in which all streams are represented by the value 1 and background data by NoData values. The reclassification was done iteratively until a sufficient amount of flow accumulation pathways were eliminated and first artefacts occurred.

The TPI, introduced by [61], can be used to determine the ruggedness of the terrain. Rough surfaces concentrate flow, promote rill formation, and consequently lead to higher sediment yields [62]. TPI is usually used for landform classifications such as hilltops, upper/mid/lower slopes, flat areas, or valleys. Its calculation as the difference between a cell value and the average cell value of the neighborhood around the cell also enables micro topographic analysis on high-resolution DEMs, and, thus, an identification of badland areas or rill systems. We calculated TPI from a smoothed DEM to reduce the noise of micro topography, such as surface morphological changes caused by stones, or small hollows detected by high-resolution DEM.

For the final RGB+DEM map the areas classified as moderately or severely degraded based on the bare soil density were further analysed and combined with the terrain attributes. Vegetated areas with more than 50% of the area having a negative TPI and fully enclosed by a degraded LU where attributed to the corresponding degraded LU. Vegetated or moderately degraded areas where more than 50% of the area had a negative TPI were attributed to the moderately degraded LU or the severely degraded LU, respectively.

## 3. Results

### 3.1. Field Mapping

A Field Map showing the results of the conventional land cover mapping is shown in Figure 3. The composition of shrub species differs throughout the studied area. The thorny shrub *Lycium horridum* was only found close to the former reservoir, whereas in the catchment area a mix of different shrub species occurred, these being mainly *Chrysocoma ciliata*, *Dicerothamnus rhinocerotis*, *Felicia* spp., *Helichrysum* spp., and *Selago* spp. Grasses are found throughout the whole study area, mostly interspersed with shrubs. Most of the catchment was covered with tussock grass, while short grass dominated the former reservoir area. Seven, partly bifurcated, erosion rills covering a length of 1.08 km were also identified.

### 3.2. Land Cover Classification and Accuracy

The LC classification (Figure 4B) identified 37.5% (3.64 ha) of the study area as not vegetated (bare soil or stones), 53.9% (5.25 ha) as grasses, and 8.6% (0.83 ha) as shrubs. As the FM and LC map shows, shrubs occur mainly in the northern part of the study area along the main gully and close to the former edges of the reservoir that is now fully silted-up. Additionally, they grow in higher densities alongside or within the bed of larger erosion rills. Differences in grass species, as in the field assessment, were not detected by the LC map. Non-vegetated (bare soil and stone cover) areas show some rills that were also observed on-site.

The overall accuracy of the LC classification, representing the proportion of reference sites that were mapped correctly according to the orthophoto, is 78% (Table 2). User's accuracy was highest for the bare soil (85%) and lowest for the shrubs (74%). Producer's accuracy was lowest for the grasses (66%).

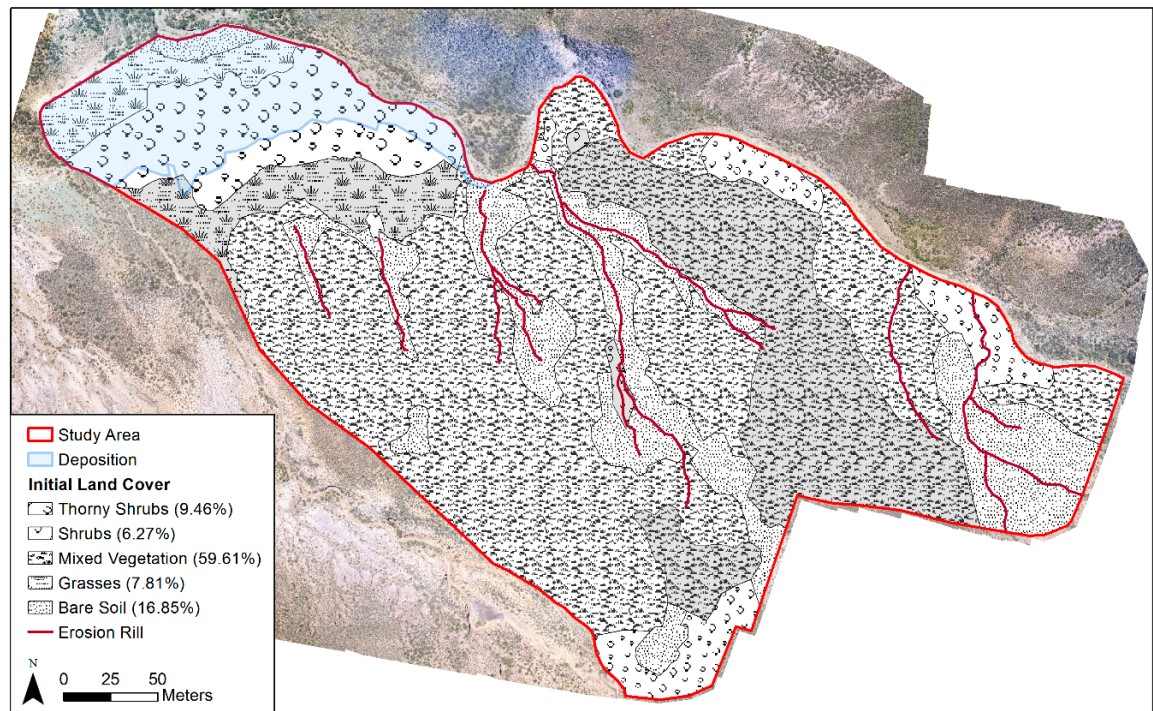

**Figure 3.** Field map (FM) derived from on-site land cover assessment. Percentages in parentheses reflect the share of the total study area. Areas highlighted in grey had a vegetation cover >75%.

**Table 2.** Confusion matrix representing accuracy assessment results for the supervised land cover classification using the SVM algorithm. Accuracy assessment was performed using 150 randomly selected sample points per LC class.

| | | Reference Data | | | | |
|---|---|---|---|---|---|---|
| | **LC Class** | **Shrubs** | **Grasses** | **Bare Soil** | **Total** | **User's Accuracy** |
| **Classified Data** | Shrubs | 111 | 37 | 2 | 150 | 74% |
| | Grasses | 20 | 114 | 16 | 150 | 76% |
| | Bare Soil | 2 | 21 | 127 | 150 | 85% |
| | Total | 133 | 172 | 145 | 450 | |
| | Producer's Accuracy | 83% | 66% | 88% | | 78% |
| | Kappa | 0.67 | | | | |

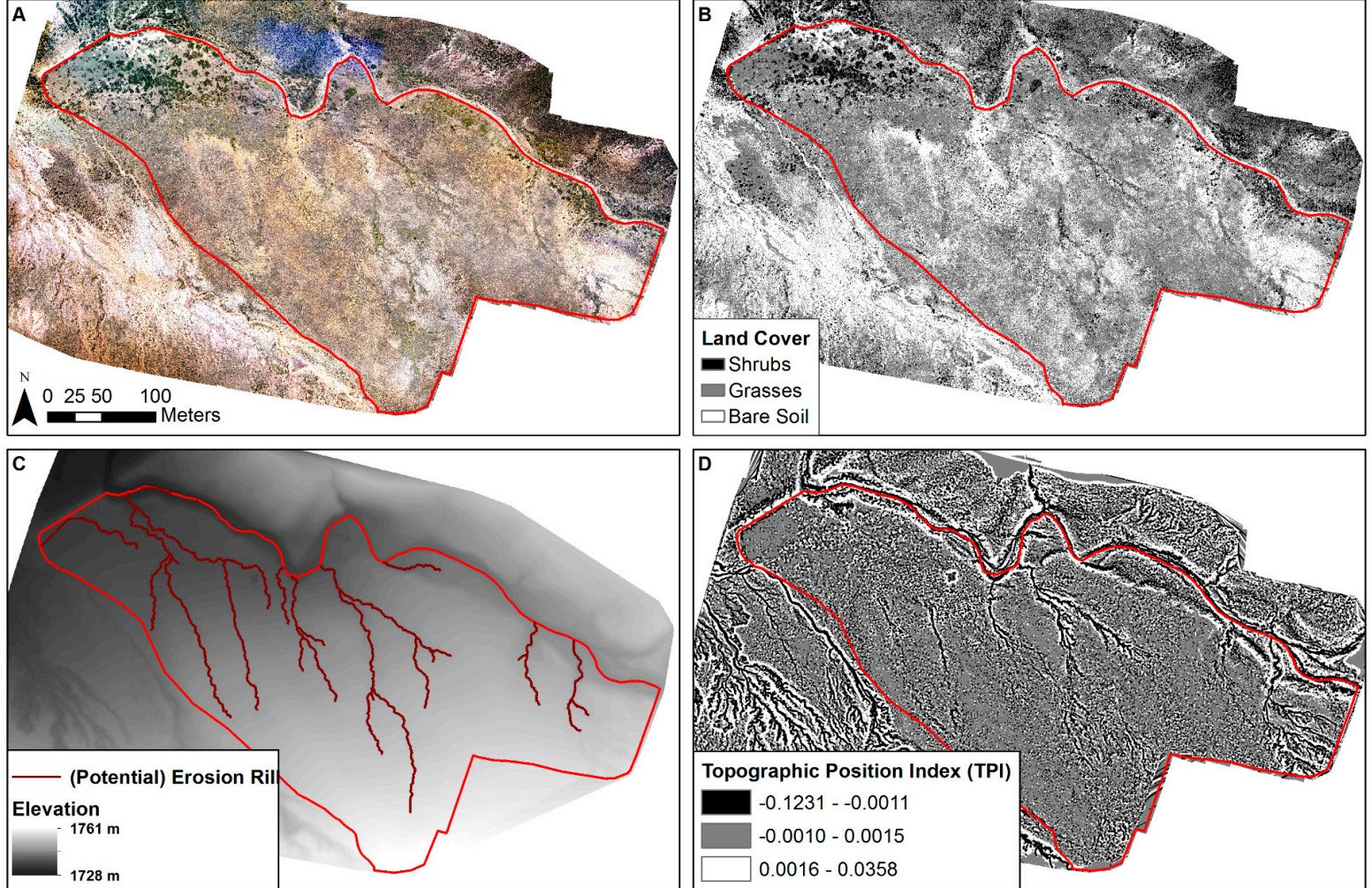

**Figure 4.** The orthophoto (**A**) was used for supervised land cover classification (**B**) (LC map). The DEM (**C**) was used for flow accumulation analysis to identify potential erosion rills and for the TPI (**D**) as a basis for landscape unit classification. The study area is marked in red. All maps are at the same scale.

### 3.3. Mapping Landscape Units

The high level of patchiness in this landscape is shown in the orthophoto (Figure 4A). Five different LUs could be identified and are described in Table 3. Visual impressions of LUs in the field and on the orthophoto are shown in Figure 5. The RGB classification (Figure 6, Table 3) attributes 0.72 ha to the severely degraded LU, 1.6 ha to the moderately degraded LU, and 6.47 ha to the vegetated LU; the RGB+DEM (Figure 7) classification accounted for 1.23 ha, 1.55 ha, and 6.15 ha, respectively. Two states of degradation could be identified: moderately and severely degraded. The latter of the two consisted of lower vegetation cover (<30%) and shrubs or grasses were smaller and sparsely distributed throughout, whereas moderately degraded areas, by contrast, were either characterized by a low vegetation cover or negative TPI. The RGB map attributed a lower amount of area to the moderately and severely degraded area (both combined made up 24%) than the RGB+DEM map (29%), and it identified fewer erosion rills (Table 3). Both the RGB and RGB+DEM maps identified all degraded areas that were mapped on the FM.

The lowest negative TPI characterizes rills or gully bottoms, which in our case can be regarded as equal to rill bottoms. Hence, erosion rills are mostly overlapping with either of the degraded LUs, or they link different areas of degradation. Most erosion rills are connected to the main gully or, as for the most rills in the west of the study area, directly to the reservoir. The area of deposition, which is characterized by the flat area behind the former dam wall, has an extent of 1.02 ha. This includes the former depositional area of the reservoir within the study area and an area severely affected by erosion located just behind the dam breach.

**Table 3.** Characteristics of landscape units for their identification in the field and on the orthophoto and their area on the classified maps. A threshold value of at least 50% vegetation cover for the vegetated LU was chosen according to the Karoo Veld assessment form in (Esler et al., 2006), where an excellent veld condition was characterized by, among other things, >50% vegetation cover. Total length of all erosion rills is given in km and marked with *.

| Landscape Unit (LU) | Identification | | Area (ha) | |
|---|---|---|---|---|
| | In Field | On UAV Imagery/DEM/Derivative | RGB Map | RGB+DEM Map |
| Depositional | Flat terrain behind the former dam wall | Former reservoir area on historical aerial images, sharp elevation change at the dam wall | 1.02 | 1.02 |
| Severely degraded | Crusted soils Contiguous area of bare soil with little vegetation (individual shrubs or tussock grass) and large gaps between individual plants, vegetation cover <30% Clearly developed rills | | 0.72 | 1.23 |
| Moderately degraded | Very low TPI Contiguous area of bare soil more regularly interspersed with vegetation but vegetation cover <50% Some rills | | 1.6 | 1.55 |
| Erosion rill | Rill structure | Line structure of bare soil, possibly flow accumulation paths on DEM Very low TPI | 0.95 * | 2.12 * |
| Vegetated | Contiguous vegetation cover interspersed with patches of bare soil or stones, at least 50% covered with vegetation | | 6.47 | 6.15 |

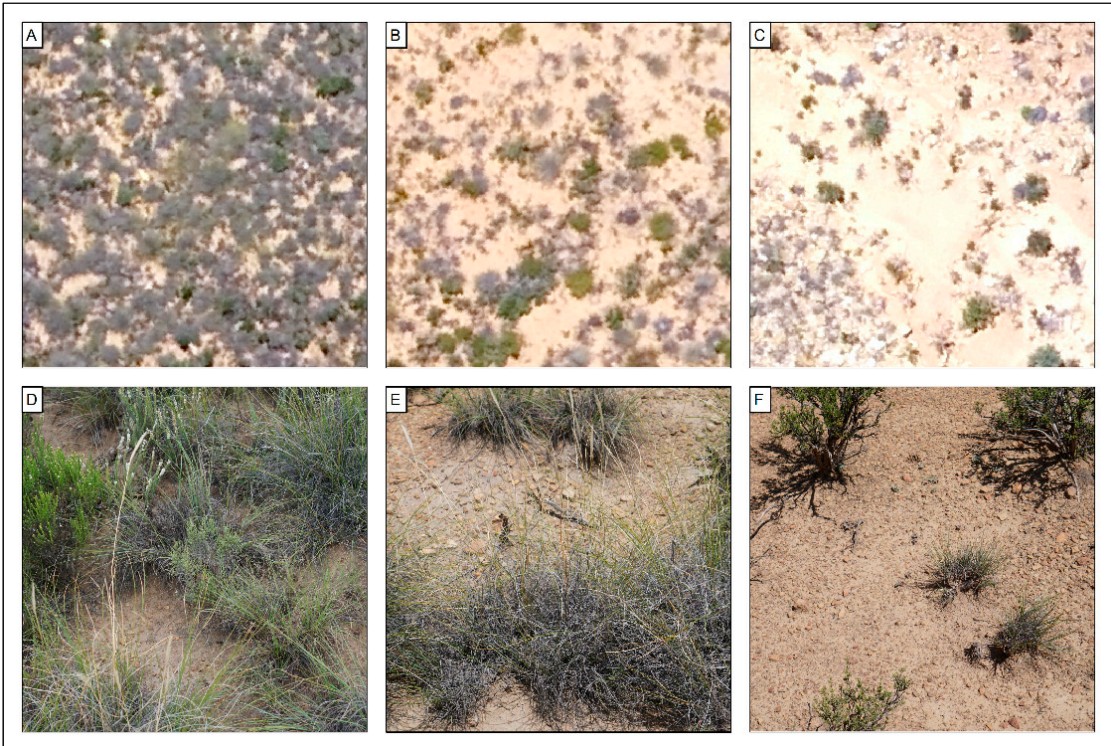

**Figure 5.** Visual appearance of different landscape units: vegetated (**A**,**D**), moderately degraded (**B**,**E**), and severely degraded (**C**,**F**). The top row (**A**–**CA; B; C;** ) shows a 10 × 10 m cutout from the orthophoto used for classification. The bottom row (**D**–**FD; E; F; A; B; C;** ) shows typical impressions from the field.

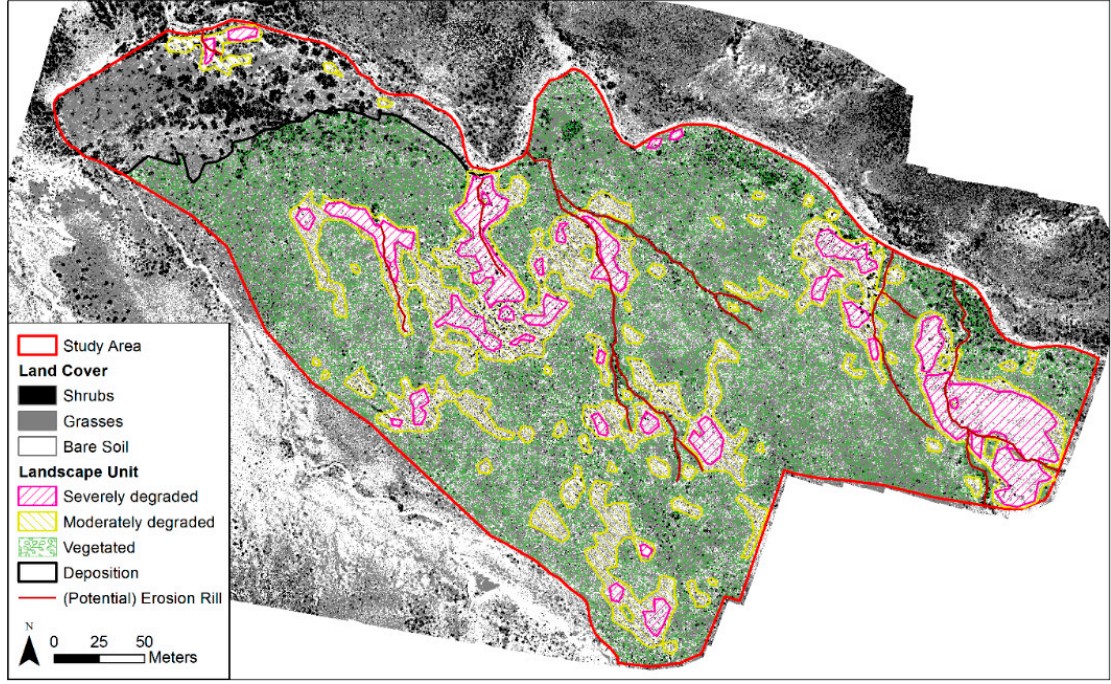

**Figure 6.** Modelled map showing identified landscape units based on classified land cover (RGB map)

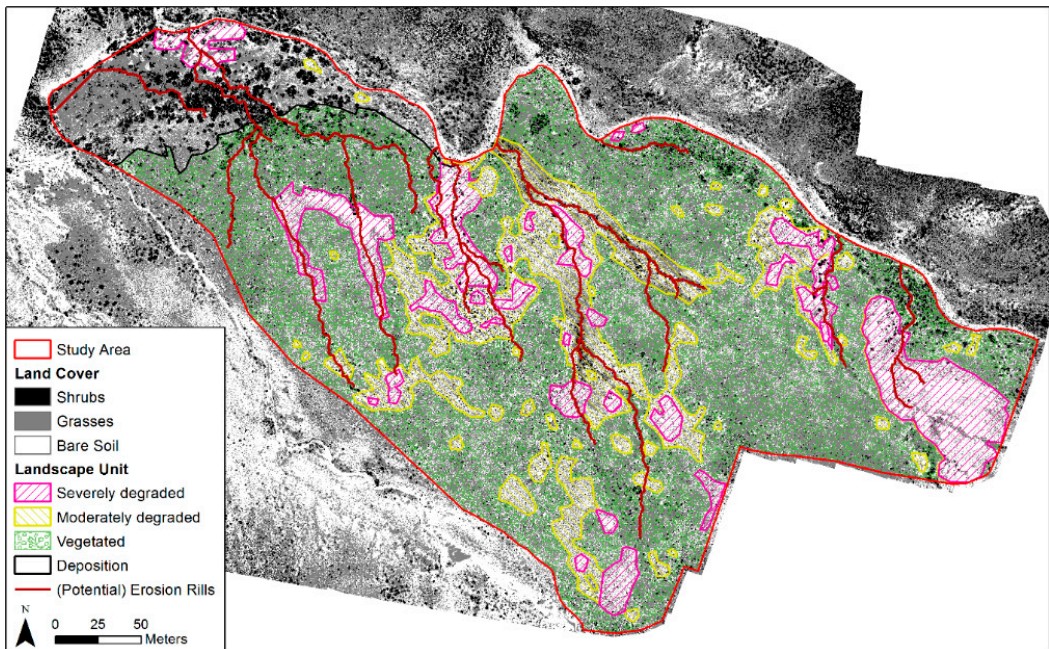

**Figure 7.** Modelled map showing identified landscape units based on classified land cover and derivatives of the digital elevation model (RGB + DEM map).

## 4. Discussion

### 4.1. Comparison of Field Mapping and RGB Map

Both methods were able to identify areas of soil degradation. However, their spatial extent differed slightly (Figure 8). Assuming that all bare soil areas of the FM are indicative of soil degradation in the UAV-derived digital RGB map, the extent of degradation mapped in the field was 5% lower than that mapped using UAV imagery. Severely degraded areas could be clearly distinguished from their surrounding by their lower vegetation cover and were identified using both approaches. However, two problems occurred. Firstly, digital mapping based on the orthophoto did not identify areas of soil degradation that had a vegetation cover larger than 50%, such as vegetated erosion rills or rills surrounded by vegetation. This lower percentage of correct identification is attributed to the vegetation cover of the rills, which leaves them partially invisible on the UAV imagery. Secondly, the UAV-based mapping revealed areas of degradation which were not identified in the field. This was probably due to a difficult delineation of cover types in the field due to (i) smooth transitions between increasing/decreasing vegetation cover or type, (ii) the spatial heterogeneity, and (iii) a limited straight down view by the surveyor as opposed to the planform overview from the UAVs' perspective. Consequently, such visual delineation of LUs in the field depends on their spatial heterogeneity and can be influenced by subjectively introducing a random mapping error, as is mentioned by [23]. In addition, smaller degraded areas that are surrounded by vegetated areas might be invisible from the surveyors' vantage point and might be missed due to time constraints during field mapping. Degraded areas that were not identified on the FM but during digital mapping are marked in white in Figure 8.

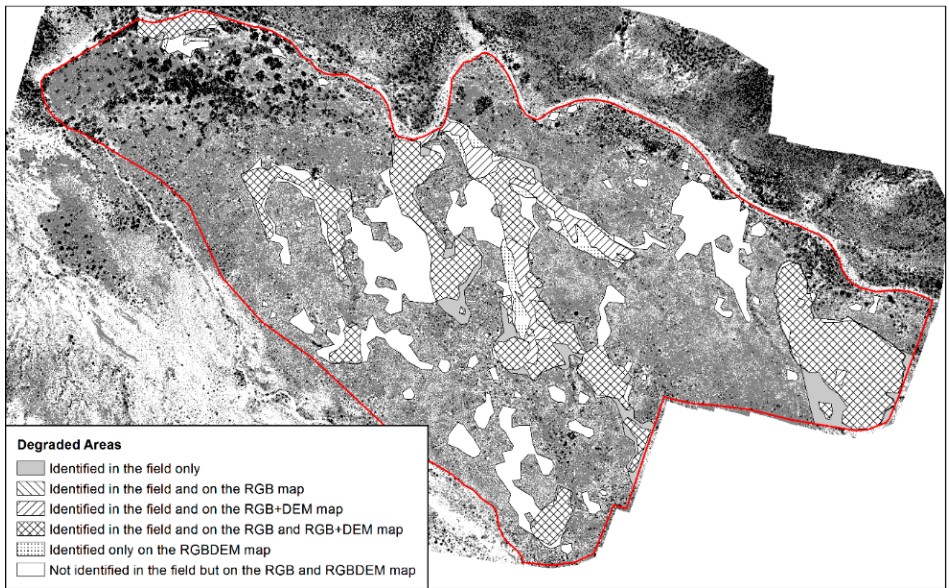

**Figure 8.** A map indicating areas that were identified as degraded with either method: manual field mapping or digital mapping.

## 4.2. Comparison of Field Mapping and RGB+DEM Map

Similarly to the digital RGB mapping, the RGB+DEM mapping also identified areas of soil degradation. The RGB+DEM map identified areas of medium vegetation cover with disturbed soil surfaces much better than the field mapping approach, leading to an additional 12% of soil assessed as moderately or severely degraded. The moderately degraded areas were largely identified with the help of the DEM and the TPI. As [57] stated in 1993, including terrain attributes and thereby increasing the information input can enhance spatial modelling. The small-scale topographic information of the roughness index (TPI) enabled the identification of areas with disturbed soil surface structure in our study area. Accordingly, small erosion rills that were not connected to larger rills were readily identifiable on the digital map. In heterogeneous landscapes, small features are more difficult to depict than large features and afford more time-investment. They can be missed by conventional mapping, illustrating a benefit of using UAVs which enable complete spatial coverage.

## 4.3. Comparison of RGB and RGB+DEM Map

Degraded, vegetated, and depositional LUs could be detected from the aerial image. However, we identified two key differences between the RGB and RGB+DEM classifications. Firstly, the RGB+DEM classification classified a larger proportion of the area as degraded, which we attribute to incorporating topography data and DEM derivates. Incorporating DEM derivates enabled the identification of moderately degraded areas that still had a high vegetation cover but already showed a high terrain roughness, which can lead to concentrated flow and rill formation if connectivity is high.

A second key difference is the greater number of rills that were identified on the RGB+DEM map than on the RGB map. In our study area, approximately 40% of the rills were covered with vegetation and thus not identified when land cover was the only parameter for identification. In particular, deeply incised rills showed some vegetation cover (Figure 9), attributed to probably a higher water availability through runoff accumulation in the rills. Vegetation in the rills also indicates that they primarily act as transport pathways for water and sediment and are not eroding actively anymore. Such erosion related features are ignored when just using RGB data. These results show that vegetation cover on aerial pictures cannot exclusively be used as a soil health indicator but emphasizes that topographic information should also be incorporated.

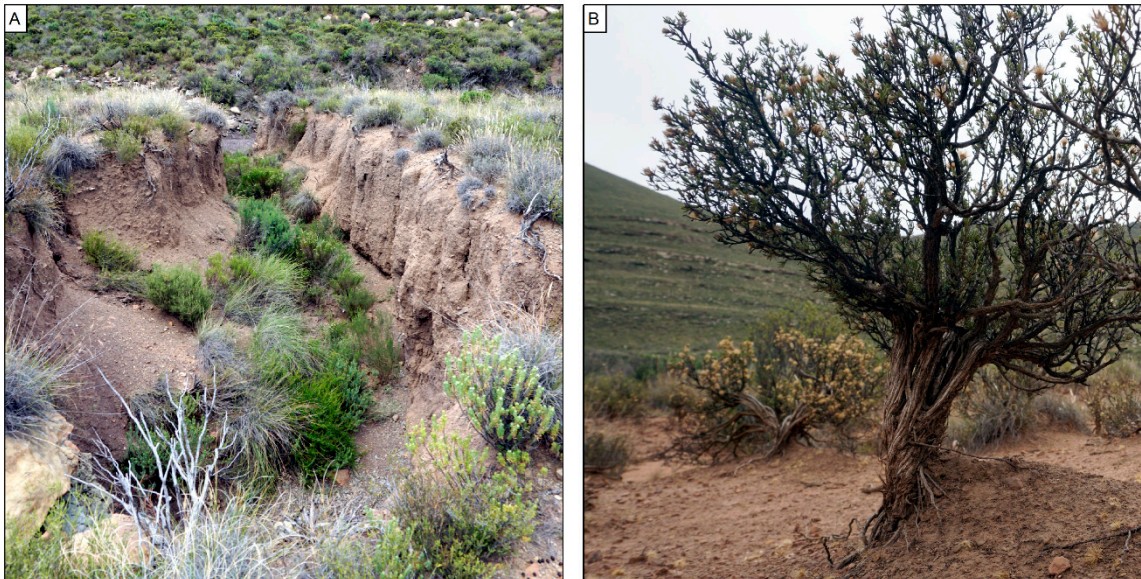

**Figure 9.** (**A**) An overgrown gully. The bottom is covered in shrubs. This will not be detected as degraded on the RGB map if vegetation cover is used only as a proxy for digital mapping. (**B**) Typical earth mound around stem/roots of shrubs in eroding areas indicate the "original" height of the soil surface. The difference between the ground to the left and right of the shrub is approximately 4 cm.

*4.4. Potential and Limitations of Using UAV Imagery for Assessing Soil Degradation*

UAV imagery served as a base for mapping landscape units. An exact classification of bare soil and vegetated areas was therefore crucial for further landscape analysis (Fiugre 4B) and degradation mapping. The kappa coefficient of 0.67 (Table 2) indicates that the classified map moderately agrees with the associated reference data [53]. It is important to emphasize that the accuracy of the classification strongly depends on the resolution of the input data used for the image analyses [25] and the selection of training data. Our training dataset used for the SVM classification was chosen very carefully. Training polygons were only assigned to a certain class if there was no doubt on the class affiliation. As a result, transitional areas between shrubs and grasses or grasses and bare soil are probably underrepresented in the training data introducing uncertainties in the distinction of boundaries between these classes. For the accuracy assessment, all randomly stratified points for the accuracy assessment could be attributed to a distinct land cover class. However, this required some ground knowledge of the differences in shapes and colours of shrubs and grasses in the study area.

Additionally, heterogeneous landscapes or land cover classes complicate a successful classification, because the landscape is composed of a mosaic of different land covers, such as shrubs, grasses, and bare soil that do not always show distinct boundaries [63]. The moderate overall accuracy of 78%, resulting in an inaccuracy of 22%, can be largely attributed to misclassified vegetation. Grass had the lowest producer's accuracy (Table 2) with two thirds of the accuracy assessment points incorrectly assigned to shrubs or bare soil. Taking a closer look at the misclassified pixels and aligning them with the field observations revealed that the edges of bushes were often misclassified as grasses (Figure 10). Additionally, misclassified shrub pixels were often accounted as *Dicerothamnus rhinocerotis*. This shrub has a dull greyish green color which was often incorrectly classified as grass. Grasses, which cover a variety of different shades of green, including very low brightness, were also misinterpreted as shrubs or bare soil (Figure 10). Using a density map to identify healthy landscape areas with a vegetation cover larger 50%, revealed many lower vegetated patches smaller 20 m$^2$ that were not detected during the field mapping. These patches could be the first indicator of arising degraded areas. However, their extent should be treated with caution due to uncertainties in boundary regions.

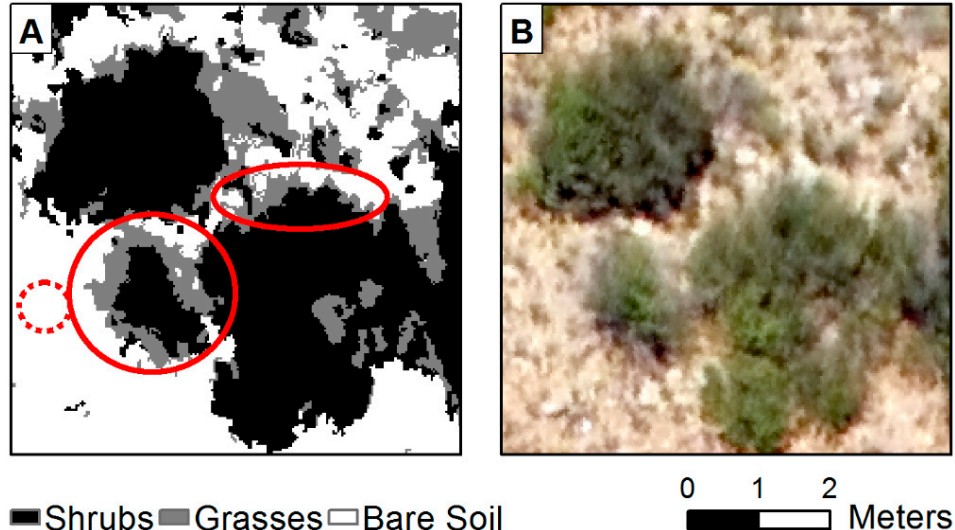

**Figure 10.** Examples of misclassification on the LC map (**A**) compared to the orthophoto (**B**): edges of shrubs were often misclassified as grasses (solid line). Small patches of grasses were incorrectly classified bare soil (dashed line).

Due to missing near-infrared spectra, which contain much information on vegetation, a separation between different shrub types and areas with mixed vegetation, as in the FM, was not successful. If more detailed information on vegetation cover and types of vegetation is needed, or a distinct separation between grasses and shrubs is necessary, multispectral data should be used [64–67]. Since the aim of our study was the mapping of soil degradation, we condensed shrub-dominated and grass-dominated areas into one LU (vegetated) and dispensed with the detailed classes as for the field mapping. Condensing these two LUs into one showed that roughly 60% of the study area is covered with vegetation. This agreed with our initial expectation based on [42] of 50–70% vegetation cover in these climatic conditions.

### 4.5. Can Field Mapping Deliver the Status Quo?

Both digitally-derived maps classified a larger area as degraded than the field mapping. However, fewer rills were visible in the RGB map than in the FM. The RGB+DEM map, using the UAV imagery and topographic data, revealed the highest percentage of degraded areas. We attribute the improved quality of the map to the identification of features such as rills which were otherwise masked by even sparse shrub cover.

The final question of our study is whether field mapping can be considered as a benchmark. Depending on the homogeneity of the landscapes, field maps can deliver a precise standard map of a study area. However, with increased heterogeneity, the time required for data collection will increase, as well as the transfer time to a geographic information system and map creation. Comparing UAV-based maps and our field map revealed that conventional mapping has further shortcomings: (i) vegetation acts as a cover and inhibits the detection of smaller patches without vegetation and soil degradation from the surveyors' position; (ii) smaller topographic features such as rills, especially if they are not connected to larger systems, might be missed because they are hidden from view by overhanging branches of shrubs; and (iii) the smooth transition between moderately degraded and non-degraded areas makes a delineation difficult in the field and introduces a subjective individual error source.

A final argument for the use of conventional UAV mapping, particularly for land management, is the fast acquisition and processing of the data compared to conventional field mapping. Even if some misclassification occurs, land cover, and implicitly soil quality changes, can be detected more regularly, which over time leads to a quality similar to or better than an out-of-date conventionally

produced map. Furthermore, the spatial resolution of UAV-derived products is higher, though their level of detail of the classification depends on input data [25] and on the spatial prediction method used [27]. However, if the vegetation type is of research interest, then a complimentary field survey is necessary, ideally also being amended with the use of multispectral data.

## 5. Conclusions

In this study, the suitability of UAV data to map soil degradation was investigated and compared to conventional field mapping. The results show that UAV imagery and data products generated with them can contribute to identifying soil degradation with similar accuracy to conventional field mapping. Most notably, UAV-based degradation mapping enabled the most severely degraded areas to be identified in a feasible manner based on their limited vegetation cover. The LC classification performed well for bare soil areas (producer's accuracy: 92%, user's accuracy: 85%), illustrating that UAV imagery delivers sufficient information for soil degradation mapping on a catchment scale. However, our results indicate that digital mapping still potentially underestimates the area of soil degradation. This may be mainly attributed to small-scale heterogeneity of land cover, a difficult distinction between bare soil ground cover and fading green tussock grasses, and shrubs inhibiting a clear view of the ground and its actual condition. Tussocks growing on pedestals or exposed shrub roots cannot be detected with UAV imagery at the catchment scale in a time-efficient way. In this case, ground-truthing would add valuable information.

Major improvements to RGB-based and vegetation density-based mapping were able to be achieved when fine-scaled topographic information derived from DEM generated from UAV data was used to identify different landscape units. Therefore, we recommend combining an orthophoto at least with a DEM and a flow accumulation model to gain information on erosion rills and moderately degraded areas.

The results of our study demonstrate that UAVs provide a valuable tool for rapid assessment of soil degradation, in particular in heterogeneous landscapes where manual field sampling is very time consuming and subject to subjective assessments by the surveyor. When exact information on vegetation cover is required and for land cover analysis in savanna-like landscapes, where RGB colors represent different shades of beige to light greyish green, multispectral cameras should be used.

**Author Contributions:** J.K. designed the study, produced the datasets, and collected the field data with P.G. and N.J.K. J.K. carried out the analysis, interpreted the results, created the figures, and wrote the manuscript. All other authors supervised the designing of the study, discussed the results, and edited the manuscript.

**Funding:** Research by the authors was funded by the University of Basel. In addition, field work by J.K. was partially funded by the Tomscik Foundation and the Freiwillige Akademische Gesellschaft Universität Basel.

**Acknowledgments:** We would like to thank Goswin Heckrath and Ruth Strunk for their assistance with fieldwork, Brigitte Kuhn for UAV piloting and helping with the post-processing, and Lena Farré for graphical advice, as well as the anonymous reviewers for their constructive review comments. The authors are grateful to Shauna Westcott and Dirk Jacobs for access to the study site.

**Conflicts of Interest:** The authors declare no conflict of interest. The founding sponsors had no role in the design of the study, in the collection, analyses, or interpretation of data, in the writing of the manuscript, and in the decision to publish the results.

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
