# Peer review of "Soil Degradation Mapping in Drylands Using Unmanned Aerial Vehicle (UAV) Data"

_soilsystems, doi:10.3390/soilsystems3020033_

Round 1
Reviewer 1 Report
This paper presents a case study using UAV imagery to assess soil degradation. It is a promising approach with broad application potentials. The paper is well written with thorough discussions. There are, however, a few major concerns regarding the methodology of the study that need to be addressed or improved.
Assessment is done based on expert knowledge only and not ground truth. In other words, all three products, the FM, RGB, and RGB+DEM maps, were not field validated. Especially for the RGB and RGB+DEM products, assessment was based on 150/LC points but the classifications was determined using expert knowledge. How accurate is this expert knowledge? There lacks validation of this. Furthermore, without any field validation, all accuracy assessments and statements were based on relative comparisons between the RGB and FM, between the RGB+DEM and FM, etc. There is no objective assessment of the results and thus methodology.
The way how the RGB+DEM map was created was not described clearly enough. It seems that there're no set criteria (fixed quantitative parameters of the input data) used to produce the map but a "manual" mapping approach as described in the paper. This introduces much subjectivity and also makes the methodology too specific for this study only and not applicable elsewhere.
When applying a SVM to classify land cover, the 40 training points/LC seem to also come from expert knowledge. Once again, the uncertainty involved in this expert knowledge was not addressed and assessed.
Author Response
Date: 19 April 2019
Responses to Reviewer #1
Reviewer Argument 1:
Assessment is done based on expert knowledge only and not ground truth. In other words, all three products, the FM, RGB, and RGB+DEM maps, were not field validated. Especially for the RGB and RGB+DEM products, assessment was based on 150/LC points but the classifications was determined using expert knowledge. How accurate is this expert knowledge? There lacks validation of this […] There is no objective assessment of the results and thus methodology.
Response:
The Field Map (FM) is only based on ground-truth data, as it is based only on our observations from the walking through the study area where we recorded major difference in vegetation cover and type as well as prominent erosion features. This was apparently not clearly described in the first manuscript. We improved the field mapping description in section 2.2. Field Mapping and hope the additional information clarifies inaccuracies.
Regarding the RGB and RGB+DEM map, our aim was to create/test a workflow based on a digital assessment because field work for a detailed erosion feature maping is very time-consuming in heterogeneous landscapes like ours. However, the underlying land cover map (classified orthophoto in bare, shrub and bare soil pixels) of both RGB and RGB+DEM map was quantitatively assessed for accuracy. We agree that there was no sufficient examination of the field knowledge used. Therefore we added extended section “4.4 Potential and limitation of using UAV imagery for soil assessing soil degradation” to elaborate on the accuracy of our methods and used knowledge. We hope this addition is a satisfactory compromise.
Reviewer Argument 2:
The way how the RGB+DEM map was created was not described clearly enough.
Response:
We improved the description for the production of the RGB+DEM map (section 2.6). Additionally we introduced now an algorithm for the detection of the vegetation/bare soil density to reduce the error of subjectivity when distinguishing LUs manually (section 2.5.). Hopefully this change in the methods makes it now also applicable elsewhere and increases objectivity of the results.
Reviewer Argument 3:
When applying a SVM to classify land cover, the 40 training points/LC seem to also come from expert knowledge. Once again, the uncertainty involved in this expert knowledge was not addressed and assessed.
Response:
We agree that there was a lack of information. Consequently we have addressed that issue in the section 4.4. The training samples were carefully chosen on areas that could be clearly assigned to one of the specific classes. The main information needed is knowledge on the typical shape and colour difference between shrubs and grasses which you automatically gain during the field visit. We hope this amendment is acceptable.

Reviewer 2 Report
L17: “a larger area as degraded than”. does not make sense. Be specific.
L18: typo!
L 27: 7 ref. is too much.
L32: first paragraph 20 ref.
Introduction need work.
L104: The field work was done across the study area? Or some samples were recorded? How many?
L121: How SVM was used? How tune the hyper parameter? How to evaluate the model?
L139: please explain more how to assign the class to the validation datasets.
L140: please add equation, also use kappa index?
L160: typo!
L219: typo!
L232: typo!
L246: ?
L321: typo! The same L344 and L349
Author Response
Date: 19 April 2019
Responses to Reviewer #2
Thank you for your review. We’ve corrected all typos, except the one in L349 because we think that the use of a comma in that sentence emphasises our point more than it would without.
In the introduction, we cut down the amount of literature to the, from our side, most relevant papers. Unfortunately the other comment on the introduction was very unspecific and did not indicate which part of the introduction should be changed.
In the methods section, we elaborated more on the performed SVM, the accuracy assessment and included the equation for the kappa index.
Please find the specific changes made below.
Comments and Suggestions for Authors
L17: “a larger area as degraded than”. does not make sense. Be specific.
à replaced by exact number “The RGB+DEM map classified 12% more as degraded than the FM”
L18: typo!
à corrected
L 27: 7 ref. is too much.
à number of references reduced
L32: first paragraph 20 ref.
à number of reference slightly reduced. However, we believe that the remaining references cited are relevant. Even though some might have the same findings, studies were conducted in various areas and show the global importance and relevance of heterogeneous patterns and their soil properties.
Introduction need work.
à comment to unspecific
L104: The field work was done across the study area? Or some samples were recorded? How many? à Yes, during field work the whole study area was covered while walking over the area and recording positions of major vegetation cover/type changes as well as prominent erosion features (such as rills, gullies). All discovered features or changes where GPS-tracked and later transformer into a map. In the revised manuscript we elaborated a bit more on the field mapping and hope the made additions are sufficient.
L121: How SVM was used? How tune the hyper parameter? How to evaluate the model?
à SVM was used with the image classification wizard in ArcGIS Pro on different segmented orthophotos. Each classified image was afterwards tested for accuracy with an accuracy assessment and an error matrix. This was part of a pre-study and is not described in detail in this paper. However, we mention the pre-study in the revised manuscript. The RGB and RGB+DEM maps were produced using the segmented orthophoto and SVM that had the highest overall accuracy.
L139: please explain more how to assign the class to the validation datasets.
à Classes for the validation dataset were assigned point by point using the orthophoto. Bare soil areas were the easiest to identify. For the validation points from the shrub and grass LC classes field knowledge on the typical shape and colour differences between shrubs and grasses in the study area was needed. We added some additional information on this process and also included a the class assignment for the validation dataset in section “4.4. Potential and limitation of using UAV imagery for soil assessing soil degradation”.
L140: please add equation, also use kappa index?
à Yes, the kappa index was used and the equation added. For a better understanding two headings (Classified and Reference Data) have been added to Table 2 in the results part.
L160: typo!
à changed from “:” to “.”
L219: typo!
à changed capital letter in L218 align both terms
L232: typo!
à corrected
L246: ?
à comment to unspecific
L321: typo! The same L344 and L349
L321 à corrected
L344 à changed from “:” to “.”
L349 à No change. We assume this is referring to the “,” in “shrubs is necessary, multispectral…”. The use of a comma emphasises our point more than it would without a comma.

Round 2
Reviewer 1 Report
Field validation is still the only final way to assess any new soil mapping methods. Considering that the current study might be constrained by resources and that the timely publication of UAV-assisted approaches is of interest to journal readers, I agree that this manuscript could be accepted for publication.